# A Hybrid Control with PID–Improved Sliding Mode for Flat-Top of Missile Electromechanical Actuator Systems

**DOI:** 10.3390/s18124449

**Published:** 2018-12-15

**Authors:** Man Zhou, Dapeng Mao, Mingyue Zhang, Lihong Guo, Mingde Gong

**Affiliations:** 1Changchun Institute of Optics, Fine Mechanics and Physics, Chinese Academy of Sciences, Changchun 130033, China; zhouman0741@163.com (M.Z.); mdp_ciomp@126.com (D.M.); guolh@ciomp.ac.cn (L.G.); 2School of Mechanical and Electronic Engineering, University of Chinese Academy of Sciences, Beijing 100039, China; 3School of Mechanical and Electronic Engineering, Jilin University, Changchun 130025, China; gmd@jlu.edu.cn

**Keywords:** electromechanical actuator, flat-top, limit cycle oscillation, chattering, PID-ISM

## Abstract

Electromechanical actuator (EMA) systems are widely employed in missiles. Due to the influence of the nonlinearities, there is a flat-top of about 64 ms when tracking the small-angle sinusoidal signals, which significantly reduces the performance of the EMA system and even causes the missile trajectory to oscillate. Aiming to solve these problems, this paper presents a hybrid control for flat-top situations. In contrast to the traditional PID or sliding mode controllers that missiles usually use, this paper utilizes improved sliding mode control based on a novel reaching law to eliminate the flat-top during the steering of the input signal, and utilizes the PID control to replace discontinuous control and improve the performance of EMA system. In addition, boundary layer and switching function are employed to solve the high-frequency chattering problem caused by traditional sliding mode control. Experiments indicate that the hybrid control can evidently reduce the flat-top time from 64 ms to 12 ms and eliminate the trajectory limit cycle oscillation. Compared with PID controllers, the proposed controller provides better performance—less chattering, less flat-top, higher precision, and no oscillation.

## 1. Introduction

The actuator system is an important operating unit in modern applications such as in autonomous vehicles, robotics, aircraft equipment and submarine operations. As electromechanical actuators (EMAs) have smaller volume and lighter weight than hydraulic actuators, EMAs have been widely used in many fields with limitations on space and weight, such as robotics, unmanned airplanes [1], and guided missiles [2]. With the rapid development of electric technology, EMA systems are broadly used to improve the maintainability and reliability of future aircraft [3].

However, the EMA system exhibits highly nonlinear properties, such as backlash, time-delay and friction. These nonlinear factors introduce several problems, such as tracking errors, limit cycles and poor stick-slip motion. In order to address these issues, various control schemes are proposed to control the position of the EMA system, including PID control algorithms [4,5,6,7],fuzzy control [8,9], intelligent algorithms [10], sliding mode control [11,12,13], the ADRC algorithm [14], robust control [15], active control [16], model-based prognostic algorithms [17], and compensation control [18,19,20]. Among these algorithms, PID is the most widely used because of its simplicity and reliability, though it suffers from poor robustness and weak anti-interference ability. ADRC can observe the “unknown disturbances” in real time and perform compensations, but it is difficult to find the optimal parameters of ADRC, furthermore, it also puts higher requirements on the MCU due to the calculation of so many parameters [21]. Intelligent control has become popular in recent years. However, intelligent controllers are seldom used in missiles or aerospace applications because of their poor portability and complicated calculations. Sliding mode control (SMC) exhibits excellent performance against uncertainties and nonlinearities. Pertaining to these advantages, the SMC is conceived as a reasonable choice for nonlinear systems and has been successfully used to estimate nonlinearities.

Chen proposed a robust sliding mode control scheme to solve a class of uncertain multi-input multi-output nonlinear systems with the unknown external disturbance, system uncertainty, and backlash-like hysteresis [22]. Chou proposed a digital signal processor (DSP)-based cross-coupled intelligent complementary sliding mode control system for synchronous control of a dual linear motor servo system [23]. In order to solve the nonlinearities in a motor-lead screw system, the fuzzy sliding mode control method was applied to the control of the position of a ball screw driven by a servomotor [24]. Yau proposed an adaptive sliding mode controller for a ball-screw-driven system, which achieved high precision and long-range positioning [25]. In order to actively suppress the vibrations of the flexible ball-screw drives, Chinedum designed a mode-compensating disturbance adaptive discrete-time sliding mode controller [26]. The dead zone characteristic as one of the most important non-smooth input nonlinearity widely existing in lots of practical systems [27,28]. Cat presented a new adaptive control algorithm for robot motion tracking problem to overcome noises and large uncertainties using integral sliding surface with a neural network [29]. Kuo proposed a novel adaptive PID with sliding mode control for the rotary inverted pendulum to achieve system robustness against parameter variations and external disturbances [30]. Rubio designed a sliding mode controller to regulate robotic arms with unknown behaviors in the dead zone and gravity, and the sliding mode strategy was employed to compensate the unknown behaviors [31]. For non-affine nonlinear uncertain systems, Xu proposed a terminal sliding mode control law based on adaptive fuzzy neural observer [32]. Bessa proposed a dynamic surface sliding mode control scheme combined with an adaptive fuzzy system, state observer and parameter estimator to estimate the uncertainty, friction and dead zone of the robot manipulator system [33]. Generally, the main difficulty in designing a fuzzy sliding mode controller for nonlinear systems is how to select the most appropriate initial value of the parameter vector. In order to deal with these types of problems, Chen designed a GA-based reference adaptive fuzzy sliding model controller for a nonlinear system [34].

However, the existing research focuses on the elimination of nonlinearities, which are suitable for tracking large-angle signals while ignoring the tracking of small-angle sinusoidal signals. With the development of high-performance aircraft, higher performance has been required of EMA systems. In particular, tracking small-angle sinusoidal responses requires higher performance. When nonlinearities exist in small-angle sinusoidal response system, the EMA system will have a long delay-time and a large flat-top in position tracking. The flat-top is mainly caused by friction and backlash. The phenomenon of flat-top occurs in the maximum position tracking, and it may introduce a large position tracking error, instability and self-excited oscillation. Besides, the flat-top has an extremely adverse effect on the missile system, which may cause the missile trajectory to oscillate. Hence, the design of such type of controller to compensate nonlinearities and eliminate the flat-top tends to be challenging but indispensable.

In order to solve these problems mentioned above, this paper proposes a hybrid controller with PID—improved sliding mode (PID-ISM) for flat-top of EMA system. As PID controller has the advantages of high control accuracy, static performance, dynamic performance and reliability, but poor robustness. SMC is computational simplicity and robustness to parameter variations. However, SMC has chattering problems, which may induce poor tracking performance and create undesirable oscillations. In contrast to traditional PID or traditional sliding mode controllers that missiles or aircraft usually use, this paper utilizes improved sliding mode control based on a novel reaching law to compensate static friction and eliminate the flat-top during the steering of the input signal, utilizes boundary layer and switching function to solve high-frequency chattering problem, and utilizes PID control to improve the performance index of EMA system.

## 2. Electromechanical Actuator System and Problem Formulation

### 2.1. The Structure of the Electromechanical Actuator 

The EMA system mainly consists of a brushless direct current (BLDC) motor, ball-screw speed reducer, encoder and potentiometer, slide, groove, crank, etc. Its structure is shown in Figure 1.

The velocity of the motor is measured by rotary encoder and return velocity to the speed regulator. The rotary encoder is fixed on the motor directly, which can avoid the errors introduced by the transmission mechanism. The rotating angle of EMA system is measured by the potentiometer. It is connected with output crank of EMA system, and return the rotation angle to the position regulator.

It is inevitable that nonlinearities exist in the transmission mechanism, such as friction and backlash. In order to find the primary source, this section analyzes the phenomenon of flat-top, and study the influence of the friction and backlash from theory and experiments.

### 2.2. Analysis of the Flat-Top of EMA System

#### 2.2.1. The Phenomena of Flat-Top

When the input signal is very small, the position output has a long delay-time. And there has a long flat-top when the direction of input signal has changed. As shown in Figure 2a, the flat top time is about 64 ms and the position tracking error is about 0.123°, while the input signal is 0.1°, 4 Hz.

This greatly reduces the performance of small-angle sine tracking and causes the limit cycle oscillation of the trajectory command system. As shown in Figure 2b, the amplitude and frequency of the limit cycle oscillation is about 0.25°, 10 Hz, which would make aircraft work abnormally. Hence, eliminating flat-top tends to be indispensable. The phenomenon of limit cycle oscillation is shown in Figure 2b.

#### 2.2.2. The Effect of Backlash

The backlash exists in transmission mechanism, such as motor and ball-screw speed reducer, ball-screw speed reducer and slide, slide and crank, crank and potentiometer, etc. When the EMA requires high tracking precision, the backlash cannot be ignored. Through the lag between encoder and potentiometer, the impact of backlash on the system can be obtained. The effect of backlash is shown in Figure 3.

As shown in Figure 3a, C is the backlash of transmission mechanism, j is the reduction ratio. $\theta_{m}$ is the rotating angle of the motor, while $\theta_{a}$ is the rotating angle of EMA system. In view of the effect of backlash, $\theta_{a}$ is not equal to $\theta_{m}/j$. The effective equation can be expressed as: (1)$$\theta_{a}=\left\{ \begin{array}{l} \begin{array}{cc} \frac{\theta_{m}}{j}-C & if\frac{\theta_{m}}{j}-\frac{\theta_{m}{}^{0}}{j}>0\;\&\frac{\theta_{m}}{j}-\theta_{a}{}^{0}>C \end{array}\\ \begin{array}{cc} \frac{\theta_{m}}{j}+C & if\frac{\theta_{m}}{j}-\frac{\theta_{m}{}^{0}}{j}<0\;\&\frac{\theta_{m}}{j}-\theta_{a}{}^{0}<-C \end{array}\\ \begin{array}{cc} \theta_{a}{}^{0} & \;else \end{array} \end{array}\right.$$

Figure 3b shows the time-delay between speed feedback and position feedback. There is a delay time between the motor and the potentiometer. From Figure 3b, it can be seen that the speed feedback reverses at 214 ms (milliseconds), the position feedback reverses at 223 ms, which delays about 9 ms. Undesired increasing of the backlash results in an increase of the flat-top.

#### 2.2.3. The Effect of Friction

According to Coulomb’s model, the dry friction can be described by a discrete mathematical model discriminating between static and dynamic friction phenomena [35]. In contrast, static friction has a greater effect on the static and dynamic performance of EMA systems when tracking small-angle sinusoidal. Since static friction has similar effects on the EMA position tracking system as with dead-zone. Therefore, the dead-zone model is used for static friction in this paper, which can greatly simplify the process of calculation. The effect of friction model is shown in Figure 4.

As shown in Figure 4a, $u_{v}$ is the output of controller, $v_{m}$ is the velocity of motor, $u_{f}$ is the starting value. The model of dead-zone can be expressed as:(2)$$v_{m}=\{\begin{array}{cc} 0 & \left|u_{v}\right|<u_{f}\\ v & u_{v}\geq u_{f}\\ -v & u_{v}\leq -u_{f} \end{array}\begin{array}{cc} , & \left(v>0\right) \end{array}$$

In Figure 4b, “speed” shows the speed dead zone. Due to static friction, the dead-zone time is approximately 57 ms.

From the analysis that mentioned above, it can be concluded that due to the static friction and backlash, there is a large flat-top when tracking the small-angle sinusoidal; friction causes speed dead-zone, and the dead-zone time is about 57 ms, accounting for about 89% of the entire flat top. Hence, compensating the static friction and reducing speed dead-zone can obviously weaken the flat-top.

## 3. Hybrid Control with PID–Improved Sliding Mode

As PID algorithms have satisfactory dynamic and static performance, they have been widely used in engineering. In addition, PID algorithms are simple and can be easily implemented in real-time processes, but they have poor robustness and low efficiency. This paper introduces a hybrid controller with PID–improved sliding mode to compensate the friction and weaken the flat-top. Some parameters are shown in Table 1.

### 3.1. The Model of Electromechanical Actuator Systems

The mathematical model of the EMA system without a controller can be written as follows:

The voltage equation of the rotor circuitry is:(3)$$u=K_{e}\omega (t)+i_{a}R_{a}+L_{a}\frac{di_{a}}{dt}$$
where u is the motor input voltage (V), $i_{a}$ is the armature current (A), $R_{a}$ is the armature resistance (Ω), $L_{a}$ is the armature inductance (H), ω is the rotor angular velocity (rad/s), $K_{e}$ is the motor electrical constant (V·s/rad).

The dynamic equation of the motor is:(4)$$\left\{ \begin{array}{l} K_{m}i_{a}=J_{m}\frac{d\omega }{dt}+T_{L}+F_{f}+b\omega \\ T_{m}=K_{m}i_{a} \end{array}\right.$$
where $K_{m}$ is motor torque constant(N·m/A),$J_{m}$ is the moment of inertia($kg\cdot m^{2}$), and $T_{L}$ is load torque(N·m). The motor torque constant and motor electrical constant are the same for an ideal motor, $T_{m}$ is electromagnetic torque, b is equivalent damping coefficient,$F_{f}$ is total friction torque.

Using Equations (3) and (4) and Laplace transform, transfer function of motor can be expressed as:(5)$$\left\{ \begin{array}{l} K_{m}I_{a}=J_{m}\omega s+T_{L}+F_{f}+b\omega \\ U(s)=K_{e}\omega (s)+I_{a}R_{a}+L_{a}I_{a}s \end{array}\right.$$

From Equation (5), the open loop transfer function of motor can be derived:(6)$$\frac{\omega (s)}{U(s)}=\frac{K_{m}I_{a}-T_{L}-F_{f}}{K_{e}(K_{m}I_{a}-T_{L}-F_{f})+(J_{m}s+b)(I_{a}R_{a}+L_{a}I_{a}s)}$$

Assuming that the load torque $T_{L}$ is zero as external disturbance, the open loop transfer function of motor can be written as follows:(7)$$\frac{\omega }{U}=\frac{1/K_{e}-F_{f}/T_{m}K_{e}}{1-F_{f}/T_{m}+\tau_{m}s+\tau_{m}\tau_{e}s^{2}}$$
where $\tau_{m}=\frac{R_{a}J_{m}}{K_{m}K_{e}}$, $\tau_{e}=\frac{L_{a}}{R_{a}}$, $\tau_{m}$ is the mechanical time constant and $\tau_{e}$ is the electric time constant.

The deceleration ratio of transmission is j, the output angle of actuator is θ.The open loop transfer function of EMA system can be written as follows:(8)$$G_{EMA}=\frac{\theta }{U}=\frac{\frac{1}{K_{e}j}-\frac{F_{f}}{T_{m}K_{e}j}}{s-\frac{F_{f}}{T_{m}}s+\tau_{m}s^{2}+\tau_{m}\tau_{e}s^{3}}$$

As $\tau_{e}$ is very small, and $\tau_{e}\ll \tau_{m}$, assuming that $\tau_{e}=0$, then:(9)$$G_{EMA}=\frac{\frac{1}{K_{e}j}-\frac{F_{f}}{T_{m}K_{e}j}}{s-\frac{F_{f}}{T_{m}}s+\tau_{m}s^{2}}$$

The inverse Laplace transform of Equation (9) is:(10)$$\ddot{\theta }=-\frac{1}{\tau_{m}}\dot{\theta }+\frac{1}{\tau_{m}K_{e}j}u+\frac{F_{f}}{T_{m}}(\frac{1}{\tau_{m}}\dot{\theta }-\frac{1}{\tau_{m}K_{e}j}u)$$

Assuming $\Delta =-\frac{F_{f}}{T_{m}},f(\theta ,t)=\frac{1}{\tau_{m}}\dot{\theta },b=\frac{1}{\tau_{m}K_{e}j}$, the model of the EMA system without controller can be written as follows:(11)$$\ddot{\theta }=-f(\theta ,t)+bu(t)\;+\;\Delta [-f(\theta ,t)+bu(t)]$$

Define $d(t)=-\frac{F_{f}}{T_{m}}[-f(\theta ,t)+bu(t)]$, then:(12)$$\ddot{\theta }=-f(\theta ,t)+bu(t)\;+\;d(t)$$
where d(t) is the disturbance.

### 3.2. PI Control

Typical EMAs utilize PID algorithm for system control. The conventional controller consists of position regulator and speed regulator. $U_{P}$ is the output of the position regulator, and it can be represented as:(13)$$U_{P}=K^{p}{}_{P}•e_{P}+K^{p}{}_{I}\int e_{P}$$
where $e_{p}=\theta_{0}-\theta$ is the position tracking error, $\theta_{0}$ is the reference angle and θ is the measured angle.

$U_{V}$ is the output of the speed regulator, and it can be represented as:(14)$$U_{V}=K^{v}{}_{P}•e_{V}+K^{v}{}_{I}\int e_{V}$$
where $e_{V}=U_{P}-v$ is the speed tracking error, and v is the measured speed.

### 3.3. Sliding Mode Control Based on Novel Reaching Law

As flat-top is mainly caused by static friction when the input is a small-angle sinusoidal, this paper utilizes SMC to compensate the static friction. In order to solve the chattering problem caused by general sliding mode control, this paper introduces the switching function and the boundary layer. Because the PI controller has advantages in stability and reliability, the PI controller is used in the boundary layer. Sliding mode control is used to improve the robustness outside the boundary layer. The structure of controller is shown in Figure 5.

#### 3.3.1. Design of Improved Sliding Controller

Define the sliding surface as follows:(15)$$s(t)=ce_{p}(t)+{\dot{e}}_{p}(t)$$
where $e_{p}=\theta_{0}-\theta$ is the position tracking error, ${\dot{e}}_{p}(t)$ is the rate of error.

In order to cancel the integration of speed loop and reduce the requirement of compensation accuracy, a new reaching law is proposed to improve efficiency. The reaching law can be introduced as:(16)$$\left\{ \begin{array}{l} \dot{s}=-\eta \mathrm{sgn}(s)\;+\;k\int e_{V}\\ \eta >0,k>0 \end{array}\right.$$

Define function sgn(s) as:(17)$$\mathrm{sgn}(s)=\{\begin{array}{lll} 1 & , & s>0\\ -1 & , & s\leq 0 \end{array}$$

If Equations (12) and (15) are substituted into Equation (16), then:(18)$$\begin{array}{l} \dot{s}=-\eta \mathrm{sgn}(s)+k\int e_{V}=c{\dot{e}}_{P}+{\ddot{e}}_{P}\\ =c{\dot{e}}_{P}+({\ddot{\theta }}_{0}-\ddot{\theta })\\ =c{\dot{e}}_{P}+{\ddot{\theta }}_{0}+f(\theta ,t)-bu(t)-d(t) \end{array}$$

The control law of sliding mode control is $U_{eq}$:(19)$$U_{eq}=\frac{1}{b}[c{\dot{e}}_{p}+{\ddot{\theta }}_{0}+\;f(\theta ,t)-d(t)\;+\;\eta \mathrm{sgn}(s)-k\int e_{v}]$$

To eliminate the chattering and reduce the accuracy requirement, the SMC works only once when the position error passes through the boundary layer.

Switching function $sat^{*}(s)$ is defined as follows:(20)$$\left\{ \begin{array}{l} sat^{*}(s)=\{\begin{array}{ccccc} \left|\mathrm{sgn}(s)\right| & , & \left|s\right|>\alpha_{0} & and & flag(k)\neq flag(k-1)\\ 0 & , & \left|s\right|\leq \alpha_{0} & or & flag(k)=flag(k-1) \end{array}\\ flag=\left\{ \begin{array}{l} 1\begin{array}{cc} , & s>\alpha_{0} \end{array}\\ -1\begin{array}{cc} , & s<-\alpha_{0} \end{array} \end{array}\right. \end{array}\right.$$
where $\alpha_{0}$ is the boundary layer, flag is the mark function.

The output of EMA controller can be written as follows:(21)$$U=U_{V}+U_{eq}•sat^{*}(s)$$

If Equations (14) and (19) are substituted into Equation (21), then:(22)$$\begin{array}{l} U=U_{V}+U_{eq}•sat^{*}(s)\\ =K^{v}{}_{P}•e_{V}+K^{v}{}_{I}\int e_{V}+\frac{1}{b}[c{\dot{e}}_{p}+{\ddot{\theta }}_{0}+\;f(\theta ,t)-d(t)\;+\;\eta \mathrm{sgn}(s)-k\int e_{v}]•sat^{*}(s) \end{array}$$

(1) When $\left|s\right|>\alpha_{0}$ and flag(k)≠flag(k−1), the controller output U can be written as:(23)$$\begin{array}{l} U=K^{v}{}_{P}•e_{V}+K^{v}{}_{I}\int e_{V}+\frac{1}{b}[c{\dot{e}}_{p}+{\ddot{\theta }}_{0}+\;f(\theta ,t)-d(t)\;+\;\eta \mathrm{sgn}(s)-k\int e_{v}]•sat^{*}(s)\\ =K^{v}{}_{P}•e_{V}+(K^{v}{}_{I}-\frac{k}{b})\int e_{V}+\frac{1}{b}c{\dot{e}}_{p}+\frac{1}{b}{\ddot{\theta }}_{0}+\frac{1}{b}f(\theta ,t)+\frac{1}{b}\eta \mathrm{sgn}(s)-\frac{1}{b}d(t) \end{array}$$

The input signal is 0.1°, and frequency is 4 Hz. When input direction changes at time k, $e_{P}(k)$, $\frac{1}{b}c{\dot{e}}_{p}(k)$ and $\frac{1}{b}{\ddot{\theta }}_{0}(k)$ are very small compared to $\frac{1}{b}\eta \mathrm{sgn}(s)$ and $\frac{1}{b}d(t)$. Due to static friction, when the input direction changes there is a large speed dead-zone,$e_{V}$ and $\dot{\theta }$ are approximately zero, as shown in Figure 4.

Then the equation above can be simplified as follows:(24)$$U=(K^{v}{}_{I}-\frac{k}{b})\int e_{V}+\frac{1}{b}\eta \mathrm{sgn}(s)-\frac{1}{b}d(t)$$

Due to the integration effect, EMAs cannot track the input well during the steering of the input signal, showing a large lag. In order to cancel the integration, we defined that:(25)$$K^{v}{}_{I}-\frac{k}{b}=0$$

Then:(26)$$U=G\mathrm{sgn}(s)-\frac{1}{b}d(t)$$
where $G=\frac{1}{b}\eta$, and Gsgn(s) is used to compensate the static friction and eliminate the flat-top.

(2) When $\left|s\right|<\alpha_{0}$ or flag(k)=flag(k−1), the controller output U can be written as:(27)$$\begin{array}{l} U=U_{V}+U_{eq}\times 0\\ =K^{v}{}_{P}•e_{V}+K^{v}{}_{I}\int e_{V} \end{array}$$

From Equations (26) and (27), because of the role of ”flag”, it can be deduced that when the position error passes through the boundary layer, the sliding mode controller will work once for each, which will reduce the time of flat-top and the influence on the key performance indexes of EMA system. Moreover, G does not require precise valve, as the sliding mode controller can only work once at a time, and the value will be corrected after 3–5 steps iterative calculation.

#### 3.3.2. The Existence and Reachability of Sliding Mode

(1) Select the Lyapunov function as:(28)$$V=\frac{1}{2}s^{2}$$

Therefore:(29)$$\dot{V}=s\dot{s}$$

From Equations (15)–(17), it can be obtained that:(30)$$\dot{V}=s\dot{s}=-s[\eta \mathrm{sgn}(s)-k\int e_{V}]$$

When:(31)$$\eta \geq \left|k\int e_{V}\right|\begin{array}{cc} , & \eta >0 \end{array}$$

We have:(32)$$-s[\eta \mathrm{sgn}(s)-k\int e_{V}]\leq 0$$

If $\eta \geq \left|k\int e_{V}\right|$, then $\dot{V}\leq 0$. Therefore, the next step is required.

(2) When the input direction changes, there is a large speed dead-zone and $e_{V}$ is approximately zero, so we have:(33)$$\begin{array}{l} \left|U\right|=\left|K^{v}{}_{P}•e_{V}+K^{v}{}_{I}\int e_{V}\right|\leq \left|U_{0}\right|\\ \Rightarrow \left|K^{v}{}_{I}\int e_{V}\right|<\left|U_{0}\right|\\ \Rightarrow k\left|\int e_{V}\right|<b\left|U_{0}\right| \end{array}$$
where $U_{0}$ is starting value of EMA system.

In order to completely overcome the friction, letting $\eta \geq b\left|U_{0}\right|$, then:(34)$$\frac{1}{b}\eta =G\geq \left|U_{0}\right|>\frac{k}{b}\left|\int e_{V}\right|$$

Then we get:(35)$$\eta >\left|k\int e_{V}\right|\Rightarrow \dot{V}<0$$

In engineering implementation, assuming $G=\left|U_{0}\right|$, so this method can assure the existence and reachability of sliding mode.

## 4. Experimental Verification

This section experimentally validates the effectiveness of the PID-ISM for the EMA system.

### 4.1. Experimental Platform

The experimental platform for verifying the performance of the PID-ISM algorithm is shown in Figure 6. The EMA system experimental platform consists of a TMS320F28335 processor, electromechanical actuator, AC-DC power, DC-DC power, CANoe, digital signal generator and a PC.

PID-ISM algorithm is implemented in the TMS320F28335 processor. A digital signal generator is used to generate position commands. The EMA system and digital signal generator use the CAN protocol. The CANoe device is mainly used to receive position feedback and position command generated by the digital signal generator. Then, CANoe device sends position feedback and position command to the PC. 

### 4.2. The Phenomenon of Flat-Top

Because EMA shave significant flat-top when the input signal is similar to sinusoidal wave and the amplitude is less than 0.1°, in this paper the input is set to0.1°/ 4Hz. The comparisons between traditional PID controller and PID–ISM controller are displayed in Figure 7 and Table 2. 

In Figure 7, “Command” is the input signal, “PID-ISM” is the position tracking by using PID-ISM controller, and “PID” is the position tracking by using PID controller. From Figure 7a, the input signal reverses at 319 ms, and PID-ISM can significantly reduce the time of flat-top from 64 ms to 12 ms compared to the PID controller. The position output of EMA system can track the input signal accurately. Compared with the PID controller, PID-ISM reduces the position tracking error from 0.123°to 0.029°, which greatly improves the position tracking precision, as shown in Figure 7b. Figure 7c shows the output of controllers, and it can be found that PID-ISM output U reverses at 321 ms, while PID output U reverses at 355 ms. Compared with PID controller, PID–ISM controller reduces about 34 ms, and has higher efficiency. Figure 7d shows speed feedbacks, PID controller has an obvious dead zone during the steering of the input signal. The dead time is about 57 ms by using PID controller, while PID-ISM is about 10 ms.

Table 2 shows the comparison between PID and PID-ISM. From these experimental results, it can be concluded that by using the PID-ISM controller, the speed dead-zone reduce from 57 ms to 10 ms and the flat-top time reduces from 64 ms to 12 ms. At the same time, position tracking error reduces from 0.123°to 0.029°. The experimental results show that PID-ISM can significantly reduce the flat-top time and increase position precision, which is helpful to eliminate the limit cycle oscillation.

### 4.3. The Influence of PID-ISM Controller on Other Indexes

In order to analyze the influence of PID-ISM on the key performance indexes of EMA systems, such as bandwidth, position tracking precision, setting time, steady-state error and overshoot, this section presents the comparisons and experiments of the two controllers. Figure 8 and Table 3 compare the static and dynamic performance between PID and PID-ISM.

In Figure 8, “Command” is the input signal, “PID-ISM” and “PID” are the position tracking by using PID-ISM and PID algorithms. From Figure 8a,b, it can be concluded that PID-ISM controller has the same overshoot, rise time and steady-state error as the traditional PID controller and they show little difference in tracing a sinusoidal signal when the amplitude is large, as shown in Figure 8c. In Figure 8d, the input signal is 1.5°/25Hz, and the position feedback using PID-ISM controller are similar to those if the PID controller. 

The dynamic response indexes between PID-ISM and PID are shown in Table 3. From Table 3, it can be concluded that the overshoot is 8.8%, the rising time is 0.036 s, the setting time is 0.241s, the steady-state error is 0.002°and the bandwidth is higher than 25 Hz by using the traditional PID algorithm. PID-ISM algorithm has similar performance indexes with PID algorithm. The comparative results demonstrate that PID-ISM algorithm has little influence on the key performance indexes of EMA system, and solves the chattering problem caused by general sliding mode control.

### 4.4. The Limit Cycle Oscillation Of Missile Trajectory

The flat-top of position tracking has a great negative impact on the missile trajectory, such as limit cycle oscillation. In order to verify the performance of EMAs in the closed loop of missile system, EMAs are substituted into the semi-physical simulation of missiles. The results are shown as Figure 9. The “PID-ISM” and “PID” are missile trajectories using PID-ISM controller and PID controller. 

From Figure 9, there is a serious limit cycle oscillation in missile trajectory by using PID controller. The amplitude and frequency are about 0.25°, 10 Hz, which will make missile work abnormally. In contrast to the PID controller, there is almost no oscillation in missile trajectory by using PID-ISM controller. The experimental results show that the PID-ISM is helpful to eliminate the limit cycle oscillation. These experiments show that the PID-ISM controller can compensate the static friction and significantly reduce the flat-top time. In addition, PID-ISM controller can eliminate the limit cycle oscillation of the trajectory and has little influence on other key performance indexes of EMA system.

## 5. Conclusions 

Due to nonlinearities, there is flat-top phenomenon in position tracking. The flat top introduces a large position tracking error, during the steering of the input signal. It is possible to make the system self-excited oscillate, and even lead to limit cycle oscillation of the missile trajectory.

In order to compensate the static friction and reduce the flat-top time, this paper provides a hybrid control with PID–improved sliding mode controller. The proposed method utilizes sliding mode control to compensate the static friction and eliminate the flat-top, utilizes the PID control to replace the discontinuous control and improve the performance of EMA system. In order to improve efficiency, a novel reaching law is designed to cancel the integration during the steering. In addition, this paper designs boundary layer and switching functions. When the position errors pass through the boundary layer, the sliding mode controller works once for each, which can solve the chattering problem and reduce the accuracy requirements. The experiment results demonstrate that the chattering problems caused by general sliding mode have be solved. The comparisons show that PID-ISM can significantly compensate the static friction and reduce the flat-top time, which helps to eliminate the limit cycle oscillation. Moreover, this algorithm does not need an accurate compensation value which simplifies debugging. In addition, the PID-ISM algorithm has little influence on other performance indexes of the EMA system, making it possible to reduce the flat-top without redesigning and re-debugging the PID parameters. The results demonstrate the feasibility and application potential of PID-ISM algorithm. Of course, the tuning of PID-SMC is a complex process. The constraints between the parameters are not clear enough, therefore, it is difficult to find the optimal parameters of PID-SMC. Furthermore, it also puts higher requirements on the engineering experience due to the design of the boundary layer and switching functions.

## Figures and Tables

**Figure 1 sensors-18-04449-f001:**
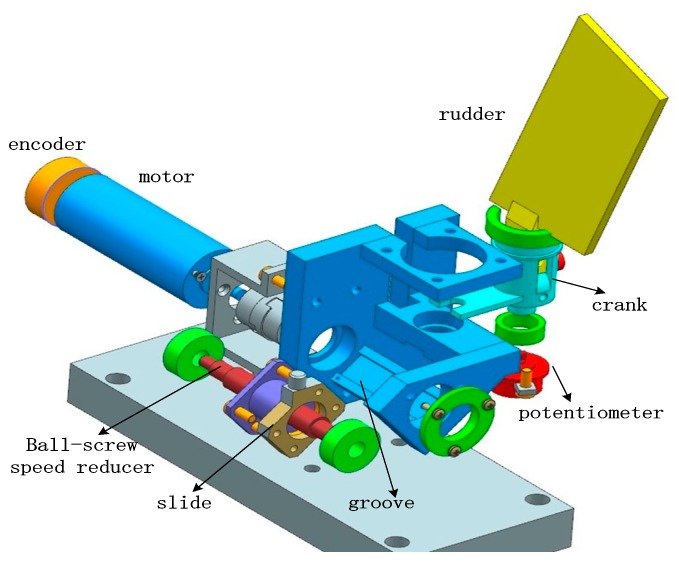
The structure of the electromechanical actuator systems.

**Figure 2 sensors-18-04449-f002:**
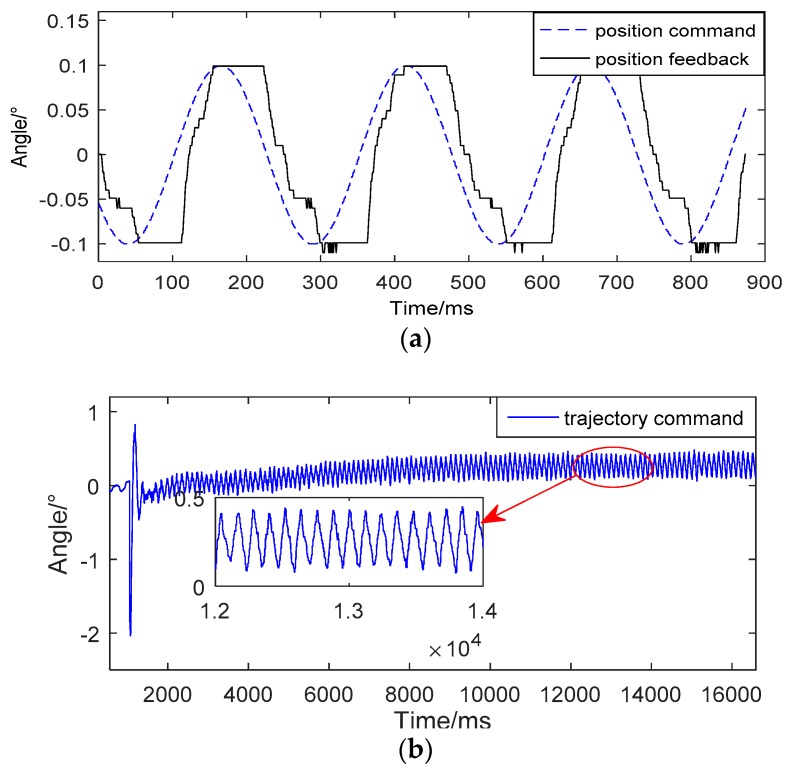
(**a**) The phenomena of flat top. (**b**) The phenomena of limit cycle oscillation.

**Figure 3 sensors-18-04449-f003:**
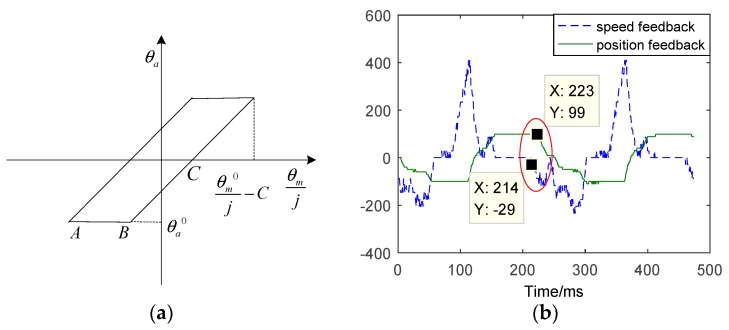
The effect of backlash. (**a**) Characteristic of backlash. (**b**) The tracking results of time-delay between speed feedback and position feedback.

**Figure 4 sensors-18-04449-f004:**
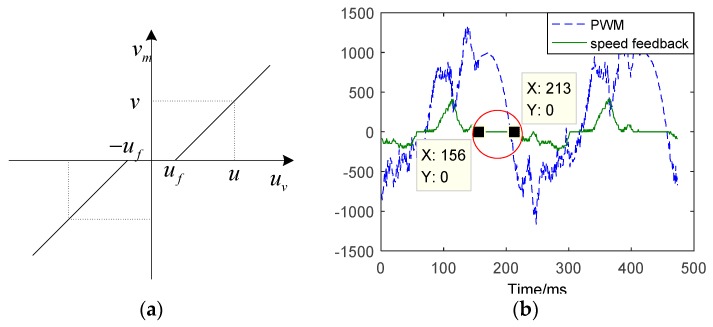
The effect of friction. (**a**) Characteristic of friction. (**b**) The tracking results of time-delay between speed feedback and PWM.

**Figure 5 sensors-18-04449-f005:**
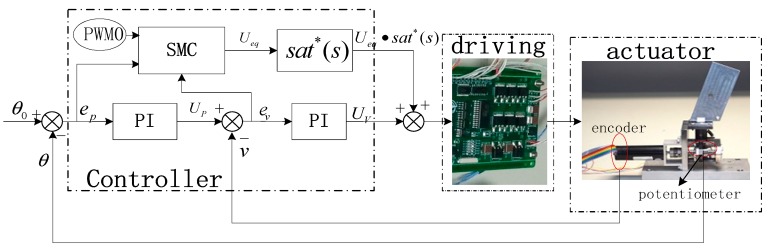
The controller of EMA system.

**Figure 6 sensors-18-04449-f006:**
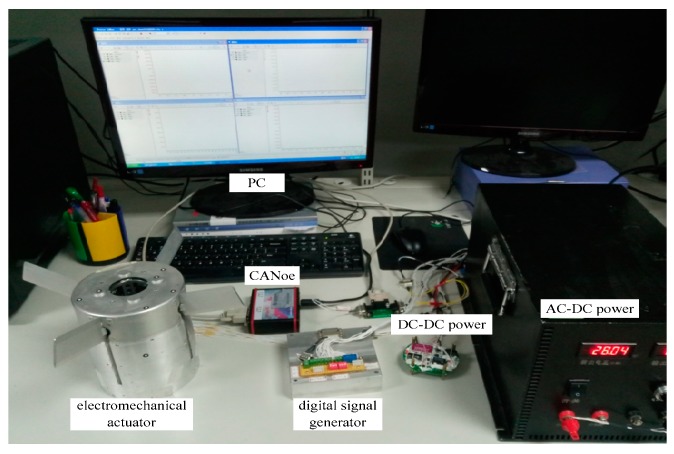
The experimental setup of EMA system.

**Figure 7 sensors-18-04449-f007:**
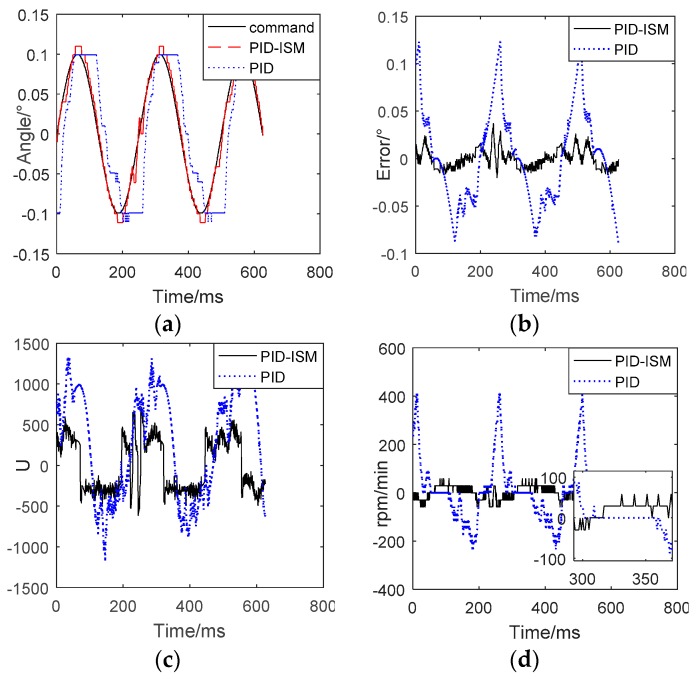
Position tracking (0.1°/4Hz) by using PID-ISM and PID. (**a**) Flat-top of position tracking. (**b**) Errors of position tracking. (**c**) The controller output of PID-ISM and PID. (**d**) The speed feedback by using PID-ISM and PID.

**Figure 8 sensors-18-04449-f008:**
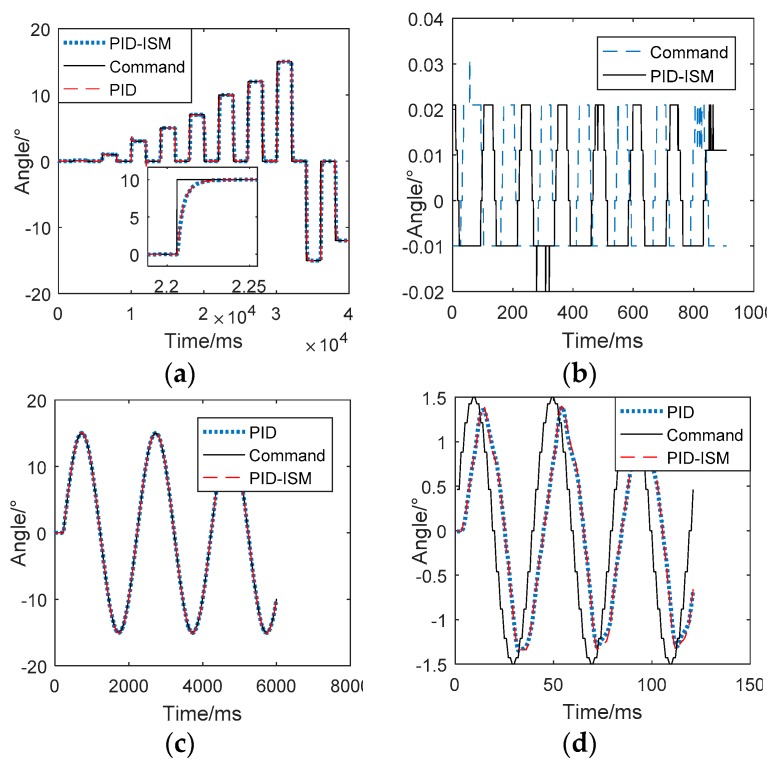
The comparisons of traditional PID and PID-ISM. (**a**) Step response. (**b**) Steady-state errors. (**c**) Position tracking by using PID and PID-ISM. (**d**) Thebandwidth test.

**Figure 9 sensors-18-04449-f009:**
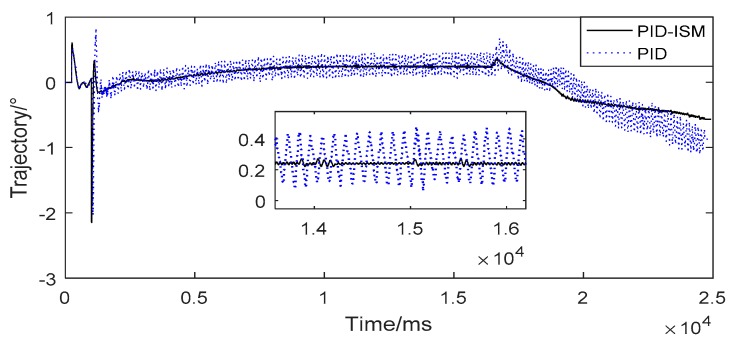
The limit cycle oscillation of missile trajectory.

**Table 1 sensors-18-04449-t001:** Parameters value of EMA system.

Symbol	Quantity
u	Motor input voltage(V)
$i_{a}$	Armature current (A)
$R_{a}$	Armature resistance (Ω)
$L_{a}$	Armature inductance (H)
ω	Rotor angular velocity (rad/s)
$K_{e}$	Motor electrical constant (V·s/rad)
$K_{m}$	Motor torque constant (N·m/A)
$J_{m}$	Moment of inertia ($kg\cdot m^{2}$)
$T_{L}$	Load torque (N·m)
$T_{m}$	Electromagnetic torque (N·m)
$F_{f}$	Total friction torque (N·m)
$\tau_{m}$	Mechanical time constant
$\tau_{e}$	Electric time constant
j	Decelerate ratio of transmission
θ	Output angle of actuator (°)
$\theta_{0}$	Reference angle (°)
d(t)	Disturbance
$e_{p}$	Position tracking error (°)
$e_{V}$	Speed tracking error (°/s)
${\dot{e}}_{p}(t)$	Rate of error (°/s)
$U_{V}$	Output of the speed regulator
$U_{P}$	Output of the position regulator
$U_{eq}$	Control law of sliding mode control
$U_{0}$	Starting value of EMA system

**Table 2 sensors-18-04449-t002:** Comparisons between traditional PID and PID-ISM.

	PID	PID-ISM	Improvement
Flat-top time (s)	0.064	0.012	81.25%
Position error (deg)	0.123	0.029	76.42%
Delay time of backlash (s)	0.009	0.006	33.33%
Speed dead zone(s)	0.057	0.010	82.46%

**Table 3 sensors-18-04449-t003:** Dynamic Response Index.

	PID	PID-ISM	Improvement
Overshoot	8.8%	9%	−2.3%
Rising time (s)	0.036	0.037	−2.8%
Setting time (s)	0.241	0.256	−6.2%
Static error (deg)	±0.003	±0.002	33.3%
Bandwidth (Hz)	≥25	≥25	0%

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
