# Peer review of "A Hybrid Control with PID–Improved Sliding Mode for Flat-Top of Missile Electromechanical Actuator Systems"

_sensors, 2018, doi:10.3390/s18124449_

Round 1
Reviewer 1 Report
General Comments
In my opinion, the paper is well written, clear and interesting; the proposed issues are fairly clear and well described, the technical-scientific problem is clearly framed and supported by a good physical-mathematical formulation and the numerical results are validated by an experimental test-bench.
However, since I think that some minor corrections are required, I suggest accepting it after MINOR REVISION.
Advantage & Disadvantage
STRENGTHS:
The paper analyses the considered topic in a proper and exhaustive way, proposing a new control approach, properly supported by numerical models and experimental evidence. As regards the scientific and technical issues considered in this paper, I do not detect significant errors, shortcomings or criticality.
WEAKNESSES:
In my opinion, I do not see significant errors or criticality in the text; nevertheless, before publishing this article, I think it is necessary to evaluate the following issues:
1) First of all, it is necessary to revise the paper text in order to correct some typos and mistakes.
In this regard, the authors can refer to the PDF file attached to this revision (entitled "Sensors-395805-peer-review-v1.pdf"), in which I highlighted in yellow someone of these errors (but not necessarily all).
2) Authors should introduce, already at the beginning of the paper, what they mean by the term "flat-top" and what are its physical causes.
3) In paragraph “1. Introduction” authors write: “In addition, hydraulic actuators in aircraft systems are high-maintenance and more vulnerable to high temperatures and pressures”. In my opinion, this sentence should be reviewed because its syntax does not convince me; furthermore, it would be opportune for the authors to justify this statement with some bibliographical reference.
4) The authors should explain what is the physical quantity plotted in figure 2 (b) and, perhaps, explicitly explain it by inserting a legend in the figure.
5) The sentence “Friction mainly contains static friction and dynamic friction” reported at the beginning of section “2.2.2. The effect of friction” (page 5) should be revised; for instance, authors could rewrite it as follow: "According to Coulomb's model, the dry friction can be described by a discrete mathematical model discriminating between static and dynamic friction phenomena".
In addition, authors could add a bibliographical reference to justify it (e.g. L. Borello, M.D.L. Dalla Vedova, "A dry friction model and robust computational algorithm for reversible or irreversible motion transmission". International Journal of Mechanics and Control (JoMaC), Vol. 13, No. 2, December 2012, pp. 37-48, ISSN: 1590-8844).
6) The authors should carefully review the equations (13) and (14) to standardize the respective notations together and make them consistent with the one shown in Figure 5.
7) As regards the equation (15), authors should explicit what is the meaning of the derivative term reported in this equation.
8) As regards the reference number 17, it should be revised in order to report correctly the corresponding authors' names: the right name of the first author is "Dalla Vedova M D L" (instead of "Vedova M D L D").
9) In conclusion, I think that a summary table explaining all the symbols and acronyms used in this paper would be very useful for its comprehension.

Author Response
Special thanks to you for your good comments. Those comments are all valuable and very helpful for revising and improving our paper, as well as the important guiding significance to our researches. We have studied comments carefully and have made correction which we hope meet with approval.
Once again, thank you very much for your comments and suggestions.

Reviewer 2 Report
This is a good paper but requires the following reviews before is accepted:
There are several papers describing the handling of similar nonlinearities in a motor-lead screw system. The literature review needs to be expanded to illustrate the novelty of the proposed approach.
Similarly, there are other papers that combine PID with sliding mode control. The paper needs to include more literature review to show the novelty of the proposed approach.
English is good but several small corrections are needed ("limit circle" should be "limit cycle", "wildly" should be "widely", and several others.
Author Response
Special thanks to you for your good comments. Those comments are all valuable and very helpful for revising and improving our paper, as well as the important guiding significance to our researches. We have studied comments carefully and have made correction which we hope meet with approval.
A revised manuscript with the correction sections highlighted was attached and for easy check/editing purpose. The main corrections in the paper and the responds to reviewer’s comments are displayed in the PDF file.
Once again, thank you very much for your comments and suggestions.
